# Upregulation of Mitochondrial Sirt3 and Alleviation of the Inflammatory Phenotype in Macrophages by Estrogen

**DOI:** 10.3390/cells13171420

**Published:** 2024-08-25

**Authors:** Maria Luisa Barcena, Céline Christiansen-Mensch, Muhammad Aslam, Natalie Haritonow, Yury Ladilov, Vera Regitz-Zagrosek

**Affiliations:** 1Department of Urology, Eberhard Karl University of Tuebingen, 72076 Tuebingen, Germany; 2German Center for Cardiovascular Research (DZHK), Berlin Partner Site, 10115 Berlin, Germany; 3Institute for Gender in Medicine, Charité–Universitätsmedizin Berlin, 10115 Berlin, Germany; christiansen@ascenion.de (C.C.-M.);; 4Experimental Cardiology, Department of Internal Medicine I, Justus Liebig University, 35392 Giessen, Germany; muhammad.aslam@physiologie.med.uni-giessen.de; 5German Center for Cardiovascular Research (DZHK), RheinMain Partner Site, 61231 Bad Nauheim, Germany; 6Department of Geriatrics and Medical Gerontology, Charité–Universitätsmedizin Berlin, 12203 Berlin, Germany; natalie.haritonow@charite.de; 7Department of Cardiovascular Surgery, Heart Center Brandenburg, Brandenburg Medical School, Bernau bei Berlin, 16321 Brandenburg, Germany; yury.ladilov@rub.de; 8Department of Cardiology, University Hospital Zürich, University of Zürich, 8091 Zürich, Switzerland

**Keywords:** Sirt3, mitochondrial protein acetylation, macrophages, estradiol, sex differences

## Abstract

Background: Aging and comorbidities like type 2 diabetes and obesity contribute to the development of chronic systemic inflammation, which impacts the development of heart failure and vascular disease. Increasing evidence suggests a role of pro-inflammatory M1 macrophages in chronic inflammation. A shift of metabolism from mitochondrial oxidation to glycolysis is essential for the activation of the pro-inflammatory M1 phenotype. Thus, reprogramming the macrophage metabolism may alleviate the pro-inflammatory phenotype and protect against cardiovascular diseases. In the present study, we hypothesized that the activation of estrogen receptors leads to the elevation of the mitochondrial deacetylase Sirt3, which supports mitochondrial function and mitigates the pro-inflammatory phenotype in macrophages. Materials and Methods: Experiments were performed using the mouse macrophage cell line RAW264.7, as well as primary male or female murine bone marrow macrophages (BMMs). Macrophages were treated for 24 h with estradiol (E2) or vehicle (dextrin). The effect of E2 on Sirt3 expression was investigated in pro-inflammatory M1, anti-inflammatory/immunoregulatory M2, and naïve M0 macrophages. Mitochondrial respiration was measured by Seahorse assay, and protein expression and acetylation were determined by western blotting. Results: E2 treatment upregulated mitochondrial Sirt3, reduced mitochondrial protein acetylation, and increased basal mitochondrial respiration in naïve RAW264.7 macrophages. Similar effects on Sirt3 expression and mitochondrial protein acetylation were observed in primary female but not in male murine BMMs. Although E2 upregulated Sirt3 in naïve M0, pro-inflammatory M1, and anti-inflammatory/immunoregulatory M2 macrophages, it reduced superoxide dismutase 2 acetylation and suppressed mitochondrial reactive oxygen species formation only in pro-inflammatory M1 macrophages. E2 alleviated the pro-inflammatory phenotype in M1 RAW264.7 cells. Conclusions: The study suggests that E2 treatment upregulates Sirt3 expression in macrophages. In primary BMMs, female-specific Sirt3 upregulation was observed. The Sirt3 upregulation was accompanied by mitochondrial protein deacetylation and the alleviation of the oxidative and pro-inflammatory phenotype in M1 macrophages. Thus, the E2–Sirt3 axis might be used in a therapeutic strategy to fight chronic systemic inflammation and prevent the development of inflammation-linked diseases.

## 1. Introduction

Macrophages play a crucial role in the innate immune system and can take on different activation states based on the signals they receive. They can differentiate into M1 macrophages, which have a pro-inflammatory profile, or into M2 macrophages, which participate in anti-inflammatory responses, promote wound healing, assist in tissue remodeling, and regulate immune functions [1,2,3]. In particular, interferon-γ (IFN-γ), in combination with Toll-like receptor 4 (TLR4) ligands such as lipopolysaccharide (LPS), induce the activation of macrophages into classical activated macrophages (pro-inflammatory M1 phenotype) characterized by the release of prominent pro-inflammatory cytokines (e.g., tumor necrosis factor (TNF-α), interleukin (IL)-1β, IL-12, and IL-18, or monocyte chemoattractant protein-1 (MCP-1)) and reactive oxygen species (ROS) [4]. However, the sustained and excessive activation of the pro-inflammatory phenotype may lead to chronic inflammation and tissue damage culminating in various diseases. Furthermore, a recent study by Seegren et al. [5] suggests a role of macrophages in chronic age-associated inflammation, i.e., in inflammaging. 

Mitochondria contribute to the inflammatory response by releasing ROS [6], promoting the activation of the NLRP3 inflammasome [7], and initiating the cGAS-STING pathway [8]. Moreover, recent studies indicate that a shift in mitochondrial metabolism is essential for the macrophage activation to the M1 or M2 states, comprising alterations in oxidative metabolism, mitochondrial ROS formation, tricarboxylic acid cycle activity, mitochondrial ultrastructure, and membrane potential (for a review, see [9]). Specifically, the stimuli promoting M1 macrophage inflammatory activation suppress mitochondrial function. Conversely, the promotion of mitochondrial function in macrophages has been suggested as a promising strategy for reprogramming the pro-inflammatory M1 phenotype into the anti-inflammatory/immune regulatory M2 phenotype and the resolution of the pro-inflammatory response [10]. 

Within various pathways supporting mitochondrial function, the role of sirtuins is of great importance. Sirtuins are a conserved family of proteins with NAD^+^-dependent deacetylase activity [11]. Among the seven sirtuin isoforms, the role of mitochondria-localized sirtuin 3 (Sirt3) in supporting mitochondrial function has been intensively investigated; via the deacetylation of key proteins involved in the tricarboxylic acid cycle, oxidative phosphorylation (OXPHO), or antioxidative defense, Sirt3 contributes to the metabolism and homeostasis of mitochondria [12]. Emerging data also suggest a beneficial role of Sirt3 in inflammation. In particular, Sirt3 attenuates palmitate-induced ROS production and inflammation in proximal tubular cells [13]. Furthermore, Sirt3 contributes to the anti-inflammatory phenotype during caloric restriction [14]. Consequently, increasing Sirt3 levels may improve mitochondrial function and diminish the inflammatory phenotype in macrophages.

Recent studies have highlighted the potential of estradiol (E2) as a means of upregulating Sirt3. Specifically, E2 treatment has been demonstrated to elevate Sirt3 levels in both seminoma cells and human umbilical vein endothelial cells (HUVECs) [15,16]. Furthermore, the anti-inflammatory effects of E2 have also been well-documented [17,18,19]. It is noteworthy that, in macrophages, E2 shortens the duration of the pro-inflammatory phase and promotes the onset of the resolution phase; i.e., E2 seems to promote M1 to M2 transition [20]. 

Whether E2 may promote Sirt3 expression or activity and thus affect mitochondrial function in macrophages remains unknown. In the present study, we aimed to explore the effects of E2 on Sirt3 expression and activity in relation to mitochondrial function, ROS formation, and inflammatory marker expression in a macrophage cell line and in mouse primary bone marrow macrophages. Our data indicate that E2 treatment leads to augmented Sirt3 expression and the deacetylation of mitochondrial proteins, along with reduced mitochondrial ROS formation and the alleviation of the pro-inflammatory phenotype.

## 2. Materials and Methods

### 2.1. Animals

Male and female C57/BL6J mice, aged 8–12 weeks, were purchased from the Forschungseinrichtung für Experimentelle Medizin (FEM), Charité—Universitätsmedizin, Berlin, Germany. All experimental procedures were performed according to the established guidelines for the care and handling of laboratory animals and were approved by the animal care committee of the Senate of Berlin, Germany (T0333/08). Mice were housed in groups of 6 animals per cage with ground corn cob bedding. The room temperature was maintained at 22 ± 0.5 °C with a relative humidity between 40% and 60%. A 12/12 h light and dark cycle was applied. Water and food were provided ad libitum.

### 2.2. Macrophage Cell Line

A male mouse macrophage cell line (RAW264.7, ATCC^®^ TIB-71 TM) was purchased from the American Type Culture Collection (Manassas, VA, USA). The cell line was maintained in DMEM supplemented with 10% fetal bovine serum (FBS), 2 mmol/L L-glutamine, 1 mmol/L sodium pyruvate, 100 U/mL penicillin, and 100 μg/mL streptomycin in a 5% CO_2_ incubator at 37 °C.

### 2.3. Bone Marrow Isolation and Differentiation of Bone-Marrow-Derived Macrophages 

Bone-marrow-derived macrophages (BMMs) were prepared as previously described [3]. Briefly, bone marrow cells were isolated from femur and tibia of mice and cultured in DMEM supplemented with 10% FBS, 20% donor horse serum, β-mercaptoethanol, 1% non-essential amino acids, 1 mmol/L sodium pyruvate, 100 U/mL penicillin, and 100 μg/mL streptomycin in the presence of 20% L929 preconditioned medium containing macrophage colony-stimulating factor (M-CSF). The bone marrow cells were grown in Teflon-coated polypropylene (polythene) bags with 10% CO_2_ at 37 °C and harvested on day 10.

We chose to utilize bone-marrow-derived macrophages (BMM) due to their superior yield and optimal homogeneity [21,22]. Additionally, it is customary to compare RAW macrophage cells with bone-marrow-derived macrophages [22].

### 2.4. HUVECs

The study conforms to the principles outlined in the ‘’Declaration of Helsinki’’. HUVECs were isolated from umbilical cords derived from normal, healthy, uncomplicated pregnancies obtained from the University Hospital of Giessen, Germany with approval from the university ethics committee (AZ132/09 and AZ 18/13) and cultured as previously described [23] in complete EC culture medium (Cat # C-22010; PromoCell, Heidelberg, Baden-Württemberg, Germany). Experiments were conducted using cells at passages 1–2. Twenty-four hours before treatment, normal FBS in the medium was replaced with charcoal-stripped FBS (Pan Biotech, Aidenbach, Bayern, Germany). 

### 2.5. Macrophage Polarization

RAW264.7 cells were differentiated into M1 macrophages by treating them with 10 ng/mL of LPS (Sigma-Aldrich, Darmstadt, Hessen, Germany) and 10 ng/mL IFN-γ (PeproTech, Darmstadt, Hessen, Germany), whereas M2 macrophages were generated using 10 ng/mL of recombinant mouse IL-4 (PeproTech, Darmstadt, Hessen, Germany) and 10 ng/mL recombinant mouse IL-13 (PeproTech, Darmstadt, Hessen, Germany) for 24 h [3].

### 2.6. Estrogen Receptor (ER) Activation

For E2 treatment, RAW264.7 cells, BMMs, and HUVECs were serum-starved for 24 h in a medium containing 2.5% charcoal-stripped FBS (Pan Biotech, Aidenbach, Bayern, Germany) and subsequently treated with water-soluble 10 nmol/L E2 (Sigma Aldrich, Darmstadt, Hessen, Germany) or 10 nmol/L dextrin (Sigma Aldrich, Darmstadt, Hessen, Germany) as vehicle control for 10 h or 24 h [3].

### 2.7. Mitochondria Isolation

Mitochondria were isolated from RAW264.7 cells and BMMs as described previously [24]. Briefly, cell membrane was disrupted using mechanical homogenization with 45 strokes in a Teflon Potter homogenizer to a cell lysis efficiency of >60%, which was confirmed by trypan blue staining. The obtained crude mitochondrial fraction was subsequently washed twice with the mitochondrial resuspension buffer (250 mM D-mannitol, 250 mM HEPES, 0.5 mM EGTA, pH 7.4). To obtain pure intra-mitochondrial proteins, the mitochondrial fraction was incubated in the buffer supplemented with 10 µg/mL trypsin for 15 min at 4 °C and then centrifuged at 12,000× *g* for 15 min at 4 °C. The purity of the mitochondrial fraction was verified by western blotting for pan-cadherin (a plasma membrane marker), TATA-binding protein (TBP, a nucleus marker), and protein disulfide isomerase (PDI, endoplasmic reticulum marker). The efficiency of the trypsin treatment was confirmed by the absence of translocase of the outer mitochondrial membrane (TOM40), a mitochondrial outer membrane marker. The integrity of the mitochondria was confirmed by the presence of heat shock protein 60 (HSP60, a mitochondrial matrix protein) in the mitochondrial fraction as well as by the absence of cytochrome c (a mitochondrial marker) in cytosolic fraction [24].

### 2.8. Immunoblotting 

For protein isolation, 1.5 × 10^6^ cells were seeded and extracted using Laemmli buffer after respective treatments. The amount of total protein was evaluated using the Pierce™ 660 nm Protein Assay according to the manufacturer’s instructions. The protein concentration was determined in a flat-bottom 96-well plate using a multiwave photometer set to 660 nm. An equal amount of protein was separated using a 10 to 12% SDS-polyacrylamide gel. Following electrophoresis, the gel was transferred to a nitrocellulose membrane (Amersham, Darmstadt, Hessen, Germany). The membrane was blocked with 5% bovine serum albumin in tris-buffered saline containing 0.1% Tween-20 at room temperature for 1 h. Subsequently, the membrane was incubated with the primary antibody at 4 °C overnight. After three washes, the membrane was then incubated with the secondary antibody for 1 h at room temperature. Primary antibodies against the following proteins were used in the study: tubulin (1:10,000, Sigma-Aldrich, Darmstadt, Hessen, Germany), Pan-cadherin (1:1000, Cell Signaling Technology, Frankfurt am Main, Hessen, Germany), Sirt 3 (1:2500, Cell Signaling Technology, Frankfurt am Main, Hessen, Germany), phosphorylated ERK1/2 (1:2500, Cell Signaling Technology, Frankfurt am Main, Hessen, Germany), ERK1/2 (1:1000, Cell Signaling Technology, Frankfurt am Main, Hessen, Germany) HSP60 (1:1000, Cell Signaling Technology, Frankfurt am Main, Hessen, Germany), PDI (1:1000, Enzo Life Science, Lörrach, Baden-Württemberg, Germany), acetylated lysine (1:500, Millipore, Darmstadt, Hessen, Germany), acetylated SOD2 (1:1000, Abcam, Cambridge, UK), SOD2 (1:2500, Millipore, Darmstadt, Hessen, Germany), cytochrome C (1:1000, Abcam, Cambridge, UK), OXPHOS, (1:2000, Abcam, Cambridge, UK), ERα (1:100, Santa Cruz Biotechnology, CA, USA), ERβ (1:200, Santa Cruz, CA, USA), GPR30 (1:500, Santa Cruz, CA, USA), TOM40 (1:1000, Santa Cruz, CA, USA), catalase (1:1000, Cell Signaling, CA, USA), and TBP (1:1000, Santa Cruz, CA, USA). Tubulin, actin, or HSP60 served as reference proteins. Immunoreactive proteins were visualized using ECL Plus (GE Healthcare, Düsseldorf, Nordrhein-Westfalen, Germany) and quantified using ImageLab software (Bio-Rad Laboratories, Feldkirchen, Bayern, Germany).

### 2.9. Flow Cytometry 

The purity of the bone-marrow-derived macrophage (BMM) population was assessed using flow cytometry analysis. A sample of freshly harvested BMMs (1 × 10^6^ cells) was processed and stained with the appropriate antibodies following the manufacturer’s protocol. Fluorescently labeled monoclonal antibodies (mAbs) that specifically target macrophage-expressed proteins were utilized for phenotypic characterization. The two-color panel included the surface antigens F4/80 (1:100, Miltenyi Biotec, Bergisch Gladbach, Nordrhein-Westfalen, Germany) and CD11b (1:100, Miltenyi Biotec, Bergisch Gladbach, Nordrhein-Westfalen, Germany) [3]. Data acquisition was performed using a MACS-Quant device (Miltenyi Biotec, Nordrhein-Westfalen, Germany) with the MACSQUANTIFY™ software (Appendix A).

### 2.10. Total ROS and Mitochondrial ROS Measurements

To assess the total ROS and mitochondrial ROS levels after a 24 h treatment with E2 or dextrin, measurements were conducted according to the manufacturer’s protocol [25]. Briefly, cells were loaded with either 10 µmol/L DCF (2′,7′-dichlorodihydrofluorescein diacetate, succinimidyl ester) for total ROS or 5 µmol/L MitoSox red for mitochondrial ROS measurement for 30 min. Subsequently, the cells were washed twice with phosphate-buffered saline containing calcium chloride (1 mmol/L) and lysed with buffer. The fluorescence intensity was analyzed by excitation at 485 ± 10 nm and emission at 530 ± 10 nm using an ELISA reader and subsequently normalized to the total protein level of the sample. 

### 2.11. ELISA

The supernatant of untreated, LPS/IFN-γ-, and IL-4/IL-13-treated cells were collected and analyzed by ELISA for TNF-α (1:200, BioLegend, Koblenz, Rheinland-Pfalz, Germany) and IL-10 (1:200, BioLegend, Koblenz, Rheinland-Pfalz, Germany) according to the antibody manufacturer’s instructions [26].

### 2.12. Seahorse Assay 

Macrophages were plated at a density of 3 × 10^4^ cells/well in a 96-well Seahorse cell culture plate (Agilent XFe96), with one well per corner of the plate containing supplemented media without cells to serve as a background control. Cells were then serum-starved and treated with E2/dextrin as described above. A Seahorse utility plate (Seahorse Bioscience, North Billerica, MA, USA) containing 200 μL of calibrant media per well, along with a Seahorse injector port and probe plate, was incubated overnight at 37 °C without CO_2_. The following day, the media in the Seahorse cell culture plate was replaced with Seahorse XF assay buffer (supplemented with 10 mmol/L glucose and 2 mmol/L glutamine) and incubated in a CO_2_-free incubator at 37 °C for one hour. The designated injector ports were filled with 25 μL of the following MitoStress Test inhibitors: 1 μmol/L oligomycin (an ATP synthase inhibitor used to calculate mitochondrial O2 consumption rate); 0.75 μmol/L carbonyl cyanide-p-trifluoromethoxyphenylhydrazone (FCCP), an ionophore that uncouples ATP synthesis from the electron transport chain to assess maximal mitochondrial oxidative phosphorylation (OXPHOS); and 0.5 μmol/L antimycin and 0.5 μmol/L rotenone (inhibitors of Complex I and III, respectively, used to determine mitochondrial and non-mitochondrial O_2_ consumption) [27]. The MitoStress Test Assay was performed for 80 min on a Seahorse XF-e96 Bioanalyzer (Agilent, North Billerica, MA, USA) following the calibration of the utility plate with the injector port plate according to the manufacturer’s instructions.

### 2.13. Statistical Analysis 

All values are expressed as the mean ± standard error (SEM), where means indicate biological replicates. GraphPad Prism 10 (GraphPad Software, San Diego, CA, USA) was used for statistical analysis. Comparisons of the means between the groups were performed using either one-way ANOVA or the Whitney test. For two-group comparisons, two-way ANOVA was performed. All *p*-values were two-sided and the statistical significance level was set at 0.05.

## 3. Results

### 3.1. Estradiol Increases Sirt3 Expression and Reduces Acetylation of Mitochondrial Proteins

To test the effects of E2 on Sirt3 expression, the mouse leukemic cell line RAW264.7 was used. These cells exhibit the expression of three main ER types (Appendix A) and are responsive to E2 treatment as demonstrated by increased ERK1/2 phosphorylation (Appendix A), a typical signature of E2 action in various cell types [28,29]. Following a 24 h treatment with 10 nmol/L E2, a significant upregulation of Sirt3 expression was observed: transcriptional after 6 h and post-translational after 24 h (Figure 1A–C). 

Since Sirt3 is a predominant deacetylase in the mitochondrial matrix [30], an effect of Sirt3 upregulation on the acetylation of mitochondrial proteins was examined. A significant reduction in total mitochondrial protein acetylation and elevation of the Sirt3 content were observed after E2 treatment (Figure 1D–F).

### 3.2. Estradiol Increases Sirt3 Expression and Mitochondrial Protein Deacetylation in Primary BMMs from Female but Not from Male Mice

To strengthen the findings, we tested the effects of E2 in primary BMMs. Similar to the response in RAW264.7 cells, the primary macrophages expressed three main ER types and showed responsiveness to E2 (Appendix A). The Western blot revealed a significantly higher Sirt3 expression in cells from female mice than from male mice under basal conditions (Figure 2A,B). Treatment with E2 for 24 h further increased mitochondrial Sirt3 expression in female- but not in male-derived cells. To determine whether the ability of E2 to enhance Sirt3 expression is specific to macrophages, female HUVECs were used: here, also, a marked elevation of Sirt3 expression was found after E2 treatment (Figure 2C–E).

To examine whether the upregulation of Sirt3 by E2 in primary BMMs may be translated to mitochondrial protein deacetylation, a western blot analysis of total acetylated lysine was carried out on purified mitochondrial fractions of BMMs. Consistent with the data obtained in RAW264.7 cells, treatment with E2 reduced the acetylation of mitochondrial proteins along with the elevation of mitochondrial Sirt3 content (Figure 2F–H). Again, the effects were observed in female-derived but not in male-derived cells. 

Previous studies suggested that E2 has a positive effect on mitochondrial respiration in several cell types [31]. However, in our study, no effects of E2 on basal or maximal respiration in BMMs were observed (Appendix A).

### 3.3. Estradiol Reduces SOD2 Acetylation and Suppresses Mitochondrial ROS Production in Pro-Inflammatory Macrophages 

We further investigated whether the differentiation of macrophages to pro- and anti-inflammatory phenotypes may affect the E2-induced Sirt3 expression. For this purpose, RAW264.7 macrophages were differentiated into M1 or M2 phenotypes. The success of differentiation was confirmed by the expression of pro- and anti-inflammatory markers (TLR4, NFκB, and RELMα) and by ROS production (Figure 3A–D). The Western blot assay showed a significant Sirt3 upregulation in M0, M1, and M2 macrophages after E2 treatment (Figure 3E), whereas no effects on Sirt1 and Sirt5 were found (Figure 3F,G).

To explore the functional significance of the E2-induced Sirt3 upregulation, mitochondrial ROS production and respiration were analyzed. Under basal conditions (dextrin treatment), total cellular (DCF fluorescence) and mitochondrial (MitoSOX fluorescence) ROS production was significantly higher in M1 macrophages than in M0 or M2 macrophages (Figure 4E,F). E2 treatment completely prevented M1-polarization-induced total and mitochondrial ROS production. In contrast, no effects of E2 on ROS in M0 or M2 macrophages were observed. In M0 macrophages, E2 increased basal respiration and reduced maximal respiration, whereas no effects were found in M1 and M2 macrophages (Appendix A). 

We also proved whether E2 may affect the expression of proteins in the mitochondrial respiratory chain. Although E2 had no effects in M1 macrophages, it significantly upregulated the expression of complexes I, IV, and V in M2 macrophages (Appendix A).

We further investigated whether the anti-oxidative effect of E2 treatment may be associated with the upregulation of anti-oxidative enzymes. However, no significant effect of E2 on SOD2 and catalase expression in M0, M1, or M2 macrophages was observed (Figure 4A,B,D). SOD2 is localized in the mitochondrial matrix and may be a target of mitochondrial Sirt3; e.g., SOD2 deacetylation can elevate its activity [32]. Indeed, the analysis of the acetylated form of SOD2 revealed a significant reduction in SOD2 acetylation in M1 macrophages after E2 treatment (Figure 4C). No E2 effects on SOD2 acetylation in M0 or M2 macrophages were found. 

The anti-oxidative effect of E2 in M1 macrophages was accompanied by an anti-inflammatory effect. Specifically, the E2 treatment of M1 macrophages reduced the release of TNF-α but increased the release of anti-inflammatory IL-10 in the cell culture medium (Figure 4G,H). These effects, taken together with the reduced ROS formation, suggest the alleviation of the pro-inflammatory phenotype by E2 in M1 macrophages. E2 treatment did not affect the expression of IL-1β and IL-18 in RAW264.7 cells.

## 4. Discussion

In the present study, we aimed to examine whether E2 treatment may upregulate Sirt3 expression in macrophages and, therefore, improve mitochondrial function as well as ameliorate the pro-inflammatory phenotype. Our findings show that (i) E2 markedly increases mitochondrial Sirt3 expression and reduces mitochondrial protein acetylation in RAW264.7 macrophages as well as in primary BMMs from female but not male mice; (ii) although E2 upregulates Sirt3 in M0, M1, and M2 macrophages, it reduces SOD2 acetylation and suppresses mitochondrial ROS formation only in pro-inflammatory M1 macrophages; and (iii) E2 alleviates the pro-inflammatory phenotype in M1 macrophages.

Mitochondria have long been considered to be important regulators of macrophage biology. In particular, the inflammatory environment has been suggested to suppress mitochondrial function in pro-inflammatory M1 macrophages and other immune cell types [33]. Conversely, mitochondrial function is important in promoting the anti-inflammatory M2 polarization of macrophages [28,34]. Therefore, restoring mitochondrial function may be a useful strategy to alleviate the pro-inflammatory M1 phenotype.

In the present study, we tested the hypothesis that Sirt3, a major mitochondria-localized deacetylase, may mitigate the pro-inflammatory phenotype in M1 macrophages. The importance of Sirt3 in macrophage biology has been underlined in several studies [35,36,37]. Liu et al. [37] reported that Sirt3 overexpression in M1-like THP-1 macrophages ameliorates the pro-inflammatory phenotype. Here, we explored E2 treatment as a potential tool with which to increase Sirt3 expression in macrophages. Two previous reports demonstrated that E2 upregulates Sirt3 in seminoma cells and HUVECs [15,16]. However, whether E2 may affect Sirt3 expression in macrophages remained unknown. Thus, using RAW264.7 macrophages and primary mouse BMMs, we showed that the 24 h E2 treatment upregulated mitochondrial Sirt3. Interestingly, in primary BMMs, the Sirt3 upregulation was observed specifically in female-mouse-derived macrophages, which also showed a higher Sirt3 expression under basal conditions compared to male macrophages. 

The E2-stimulated Sirt3 upregulation was accompanied by a reduction in total mitochondrial protein acetylation, suggesting an elevation of the total deacetylase activity in mitochondria. Additionally, in BMMs, only female-derived cells responded to E2 treatment with an enhanced deacetylation of mitochondrial proteins. Thus, E2-induced Sirt3 upregulation might be responsible for the mitochondrial protein deacetylation, which was female-sex-specific. The reason for this sex difference is unclear. One might suppose that female-derived primary BMMs were primed by circulating estrogens and, therefore, responded more strongly to E2 treatment. This idea is supported by the higher Sirt3 expression in naïve female-derived BMMs. 

In agreement with our findings, several reports demonstrated that Sirt3 overexpression reduces the acetylation of numerous mitochondrial matrix proteins and is associated with the improved mitochondrial function [36,38,39]. Similarly, in our study, basal respiration was slightly but significantly elevated in naïve M0 RAW264.7 macrophages. However, in primary naïve M0 BMMs, no effects of E2 treatment on respiration rate were found. 

We further tested whether Sirt3 upregulation is accompanied by an alleviation of the pro-inflammatory phenotype in macrophages. Similar to what was observed for undifferentiated M0 macrophages, E2 led to the upregulation of Sirt3 in M1 RAW264.7 macrophages. The pro-inflammatory M1 macrophages are characterized by reduced mitochondrial respiration, increased ROS production, and the expression of pro-inflammatory markers [40], which was also observed in our study. By analyzing the effect of E2 on these parameters, we found no effect on mitochondrial respiration; however, E2 completely abolished M1-polarization-induced ROS production, which became equal to that of naïve M0 macrophages. Likewise, E2 reduced TNF-α expression while upregulating the anti-inflammatory IL-10. These data argue for the alleviation of the pro-inflammatory phenotype in M1 macrophages by E2 treatment. In agreement with our finding, previous studies also described the anti-inflammatory action of E2 in immune cells [20,41]. Although different mechanisms have been proposed, a potential role of Sirt3 has not been considered.

Oxidative stress is an important feature of inflammation. Mitochondria-localized SOD2 [42] is a major anti-oxidative enzyme in the mitochondrial matrix. The ROS detoxification activity of SOD2 is regulated by Sirt1 via its deacetylation at lysine 68 [43]. Although reports had suggested that E2 may upregulate the activity [44] or expression [42] of SOD2, no data on the acetylation of this enzyme in response to E2 treatment were available. Therefore, we addressed this issue by analyzing SOD2 acetylation in macrophages. The Western blot analysis revealed an over 50% reduction in SOD2 acetylation by E2 treatment in pro-inflammatory M1 but not in naïve M0 or anti-inflammatory/immunoregulatory M2 macrophages. Since the expression of total SOD2 was not affected by E2, we suggest that Sirt3-induced SOD2 deacetylation may enhance its activity and diminish mitochondrial ROS formation. This idea is further supported by the previously demonstrated ability of Sirt3 to deacetylate and activate SOD2 [45]. It should be mentioned that, although SOD2 was upregulated in pro-inflammatory M1 macrophages under basal conditions, which has also been described previously [46], its activity seems to be insufficient to prevent mitochondrial superoxide anion production. The present study suggests that the upregulation of Sirt3 by E2 can ameliorate the oxidative phenotype of M1 macrophages by SOD2 deacetylation. 

Although E2 upregulated Sirt3 in all types of macrophages, SOD2 deacetylation was observed only in M1 macrophages. The cause of this phenomenon is currently unclear and could be due to differences in Sirt3 activity. Indeed, Sirt3 is an NAD+-dependent deacetylase. Among several mechanisms controlling the NAD+ concentration in the mitochondrial matrix, SLC25A51, a recently discovered mitochondrial NAD+ transporter, seems to play an important role in mammals [47,48]. Although the E2 effects on SLC25A51 expression remain unknown, the activity of the transporter is regulated by cardiolipin, probably through direct binding to the three binding motifs at the SLC25A51 [49]. Since E2 may increase the cardiolipin concentration in mammalian cells [50], one may speculate that the E2–cardiolipin–SLC25A51 axis may be responsible for the SOD2 deacetylation. Whether this mechanism or other mechanisms are responsible for the M1-macrophage-specific SOD2 deacetylation remains obscure and should be a topic of future studies.

## 5. Limitations of the Study

An important limitation of the study is the failure to confirm the causal role of Sirt3 in the anti-oxidative and anti-inflammatory effects of estradiol. We tried to address this issue by downregulating Sirt3 with shRNA. Unfortunately, we did not achieve a sufficient knockdown efficiency with either the electroporation or lipofectamine methods. 

Additionally, we have tried using the Sirt3 inhibitor LC-0296, but we did not observe the expected results in the SOD2 deacetylation. Since Sirt3 is the major deacetylase in the mitochondrial matrix with known anti-oxidative [51] and anti-inflammatory [52] effects, one may assume the role of Sirt3 in the E2 effects.

## 6. Conclusions

The findings of the present study suggest a novel strategy for the upregulation of Sirt3 in macrophages by E2 treatment. Specifically, in primary BMMs, the effect on Sirt3 expression was female-sex-specific. The Sirt3 upregulation was accompanied by mitochondrial protein deacetylation and the alleviation of the oxidative and pro-inflammatory phenotype in M1 macrophages. 

## Figures and Tables

**Figure 1 cells-13-01420-f001:**
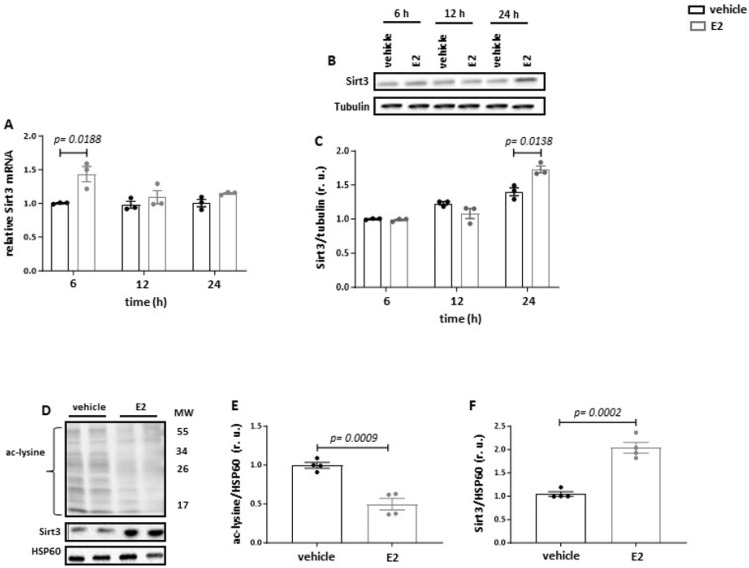
E2 increases Sirt3 expression and reduces acetylation of mitochondrial proteins in RAW264.7 cells. Real-time PCR (**A**) and western blot (**B**,**C**) analyses of Sirt3 expression performed with lysates of RAW264.7 cells treated with 10 nmol/L E2 for 6 h, 12 h, and 24 h. Western blot analyses of acetylated lysine (**D**,**E**) and Sirt3 (**F**) performed with purified mitochondrial fraction isolated from RAW264.7 cells treated with 10 nmol/L E2 for 24 h. Data are shown as means ± SEM (n = 3–4; independent experiments with technical duplicates). Data are normalized to the vehicle (at 6 h for (**A**,**C**)) and expressed in relative units (r.u.). Vehicle: dextrin; E2: estradiol; ac-lysine: acetylated lysine.

**Figure 2 cells-13-01420-f002:**
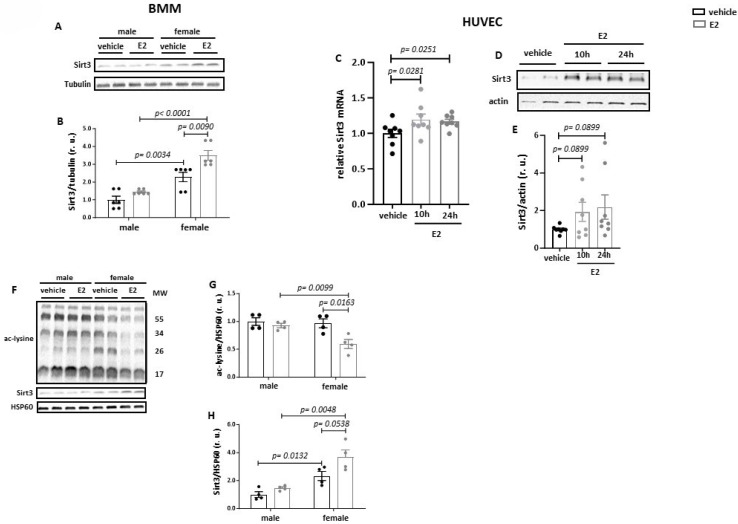
E2 increases Sirt3 expression and decreases mitochondrial protein acetylation in a sex-specific manner in BMMs and HUVECs. Western blot analysis of Sirt3 expression (**A**,**B**) performed with whole-cell lysates of male- and female-derived BMMs treated with 10 nmol/L E2 for 24 h. Real-time PCR and western blot analyses of Sirt3 performed with whole-cell lysates of female HUVECs treated with 10 nmol/L E2 for 10 h and 24 h (**C**–**E**). Western blot analyses of acetylated lysine (**F**,**G**) and Sirt3 (**H**) performed with purified mitochondrial fraction isolated from male- and female-derived BMMs treated with 10 nmol/L E2 for 24 h. Data are shown as means ± SEM (n = 4–9; independent experiments with technical duplicates). Data are normalized to the vehicle and expressed in relative units (r.u.). Vehicle: dextrin; E2: estradiol; ac-lysine: acetylated lysine.

**Figure 3 cells-13-01420-f003:**
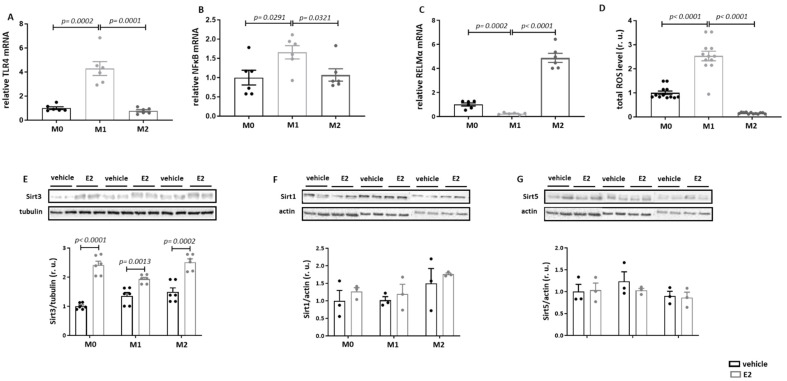
E2 elevates Sirt3 expression in naïve M0 as well as in M1 and M2 macrophages. Real-time PCR analyses of TLR4 (**A**), NFκB (**B**), and RELMα (**C**) mRNA and total cellular ROS formation (**D**) performed with lysates from naïve M0, M1 (10 ng/mL LPS and 10 ng/mL IFN-γ), and M2 (10 ng/mL IL-4 and 10 ng/mL IL-13) RAW264.7 cells treated with 10 nmol/L E2 for 24 h. (**E**–**G**) Western blot analysis of Sirt3, Sirt1, and Sirt5 expression performed with lysates of naïve M0, M1, and M2 RAW264.7 cells treated with 10 nmol/L E2 for 24 h. Data are shown as means ± SEM (n = 4–12; independent experiments with technical duplicates). Data are normalized to the M0 vehicle and expressed in relative units (r.u.). Vehicle: dextrin; E2: estradiol.

**Figure 4 cells-13-01420-f004:**
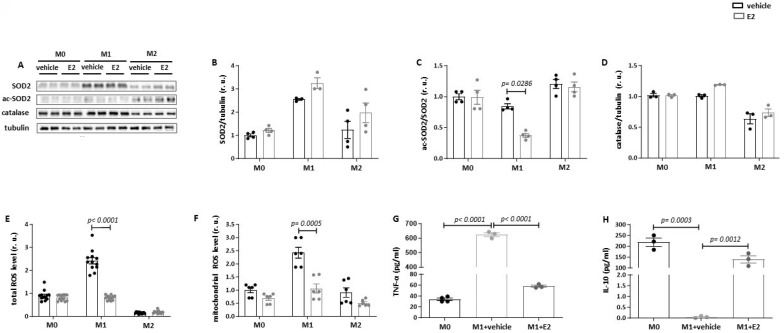
E2 leads to the deacetylation of SOD2 and decreases total and mitochondrial ROS levels in pro-inflammatory M1 macrophages. Representative images (**A**) and western blot analyses for SOD2 (**B**), acetylated SOD2 (**C**), and catalase (**D**) performed with lysates of naïve M0, M1 (10 ng/mL LPS and 10 ng/mL IFN-γ), and M2 (10 ng/mL IL-4 and 10 ng/mL IL-13) RAW264.7 cells treated with 10 nmol/L E2 for 24 h. Analysis of total (**E**) and mitochondrial (**F**) ROS formation performed with lysates of naïve M0, M1 (10 ng/mL LPS and 10 ng/mL IFN-γ), and M2 (10 ng/mL IL-4 and 10 ng/mL IL-13) RAW264.7 cells treated with 10 nmol/L E2 for 24 h. ELISA assays for TNF-α (**G**) and IL-10 (**H**) performed with cell culture medium of naïve M0, M1, and M2 RAW264.7 cells treated with 10 nmol/L E2 for 24 h. Data are shown as means ± SEM (n = 3–12; independent experiments with technical duplicates). Data are normalized to the M0 vehicle and expressed in relative units (r.u.). Vehicle: dextrin; E2: estradiol.

## Data Availability

The original contributions presented in the study are included in the article/Appendix A. Further inquiries can be directed to the corresponding author.

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
