# Peer review of "Upregulation of Mitochondrial Sirt3 and Alleviation of the Inflammatory Phenotype in Macrophages by Estrogen"

_cells, 2024, doi:10.3390/cells13171420_

Round 1

Reviewer 1 Report

Comments and Suggestions for Authors

In this manuscript the authors demonstrated that in vitro administration of estradiol (E2) upregulates Sirt3 in a macrophage cell line, which is associated to Sod2 deacetylation, reduced ROS production and blunting of proinflammatory phenotype. Although the study is innovative and interesting the following missing information/experiments must be provided.

Major points

-In the material and method section the authors describe the procedure to differentiate M1 and M2 macrophages from both RAW264.7 cell line and bone marrow. However, the majority of the experiments were performed on RAW264.7 cells and only a few on BMM naive cells. For example, the authors studied Sirt3 expression and mitochondrial protein acetylation only in BMM naive cells observing a sex-dependent response to E2. What is the effect of E2 on these parameters in M1 and M2 bone marrow derived macrophages? Is there a sex-dependent effect? Along the same line: why was the effect of E2 on ROS production, Sod2 deacetylation, oxigen consumption rate and production of proinflammatory citokines determined only in RAW264.7 cell line? The sex dependent effect of E2 on these parameters should be determined also in naive as well as in M1 and M2 male and female cells.

-How was the phenotype of bone derived primary macrophages characterized?

-Line 370-371: “However, in primary naïve M0 BMM no effects of E2 treatment on respiration rate were found” this sentence is reported in the discussion section but the relative results have not been reported. Please, amend.

-When performing the seahorse experiments, did the authors observed any effect of E2 on cytosolic utilization of glucose (glycolysis) in different kind of macrophages?

-Please add information on the software and procedure used to quantify protein content and protein acetylation in western blot experiments.

Minor points

-The OCR plot obtained with the seahorse instrument must be shown.

-Lines 270-71: “Real-Table 3. expression performed within whole-cell lysates of female 270 HUVEC treated with 10 nmol/l E2 for 10 h and 24 h” I guess the inclusion of this sentence is a mistake.

-Line 297 “e.g. SOD2 deacetylation can elevate its activity”, please add a reference

Author Response

Comments and Suggestions for Authors

Major points

-In the material and method section the authors describe the procedure to differentiate M1 and M2 macrophages from both RAW264.7 cell line and bone marrow. However, the majority of the experiments were performed on RAW264.7 cells and only a few on BMM naive cells. For example, the authors studied Sirt3 expression and mitochondrial protein acetylation only in BMM naive cells observing a sex-dependent response to E2. What is the effect of E2 on these parameters in M1 and M2 bone marrow derived macrophages? Is there a sex-dependent effect? Along the same line: why was the effect of E2 on ROS production, Sod2 deacetylation, oxigen consumption rate and production of proinflammatory citokines determined only in RAW264.7 cell line? The sex dependent effect of E2 on these parameters should be determined also in naive as well as in M1 and M2 male and female cells.

Response:

We thank the reviewer for the comments and suggestions. In the current study, the RAW264.7 cell line was used as a major biological system. Therefore, only these cells, and not BMM, were differentiated to M1 and M2 macrophages. The corresponding correction has been done in the Materials and Methods section. To prove the validity of some key results, e.g., Sirt3 upregulation, for primary cells, BMM and HUVEC have been applied (Figure. 2).

We agree that analyzing the estradiol effects on M1/M2 BMM would be interesting. However, it is currently hardly possible due to the animal experiments restriction policy in Berlin to follow the 3R- rules, i.e., replacement, reduction, and refinement, aimed to reduce the number of animals used in experiments.

-How was the phenotype of bone derived primary macrophages characterized?

Response:

The purity of the BMM population was determined via flow cytometry analysis. 1x106 cells were taken from the freshly harvested BMMs, processed, and stained with F4/80 and CD11b, as previously reported in PMID: 34867999. We present now the phenotype characterization in the supplemental figure 1.

-Line 370-371: “However, in primary naïve M0 BMM no effects of E2 treatment on respiration rate were found” this sentence is reported in the discussion section but the relative results have not been reported. Please, amend.

Response:

We apologize for this error. We addressed this issue and included the results in the supplementary figure 3. 

-When performing the seahorse experiments, did the authors observed any effect of E2 on cytosolic utilization of glucose (glycolysis) in different kind of macrophages?

Response:

The data has been accidentally deleted, and unfortunately, we no longer have access to it. We apologize for this inconvenience. 

-Please add information on the software and procedure used to quantify protein content and protein acetylation in western blot experiments.

Response:

We included the information in the material and methods section.

Minor points

-The OCR plot obtained with the seahorse instrument must be shown.

Response:

We show now a representative OCR plot in supplementary figure 4.

-Lines 270-71: “Real-Table 3. expression performed within whole-cell lysates of female 270 HUVEC treated with 10 nmol/l E2 for 10 h and 24 h” I guess the inclusion of this sentence is a mistake.

Response:

The sentence has been corrected

-Line 297 “e.g. SOD2 deacetylation can elevate its activity”, please add a reference

Response:

The reference has been included.

Reviewer 2 Report

Comments and Suggestions for Authors

Authors demonstrated the effects of E2 on the regulation of Sirt3 protein level and its deacetylation in RAW cells and BMM, moreover, E2-mediated anti-inflammatory effects on M1 and ROS. The results seem simple and clear; however, to conclude that E2-mediated Sirt3 may play a principal role in anti-inflammatory action they need more experiments to strengthen their conclusion.

Major comments

1. They should directly demonstrate whether the anti-inflammatory effects of E2 depend on Sirt3 or not by using a modality for Sirt3 knockdown or knockout. When E2-induced anti-inflammatory responses are attenuated in Sirt3 knockdown or deleted cells, they can conclude as they expect.

 2. They started their experiments based on Sirt3 upregulation by E2 alone. However,

E2 upregulated only Sirt3? Were any other Sirt proteins not influenced by E2? They need to mention something in this regard.

Minor comments

1. In Figure 1, They only demonstrated induction of Sirt3 protein at 12 h and 24. Probably, the peak of the protein induction may be between 12h and 24 h. Did they perform the time course of Sirt3 protein upregulation? If so, the peak time needs to be clarified.

 2. In addition to BMM, how about peritoneal macrophages in terms of Sirt 3 induction by E2? They need to mention why BMM was used in comparison with RAW cells.

3. In Figure 2 the legends regarding 2C-E were missing. They need to add them.

 4. As M2 polarity marker, arginase-1 or IRF4 is well known. Contrasting to RELMa, how about the marker induction by E2?

5. As shown in Figure 3F, E2-induced Sirt 3 upregulation was confirmed in each M0, M1, and M2 type RAW cell; however, deacetylation was attenuated in M1 alone. They need to discuss this phenomenon.

Author Response

Comments and Suggestions for Authors

Major comments

  1. They should directly demonstrate whether the anti-inflammatory effects of E2 depend on Sirt3 or not by using a modality for Sirt3 knockdown or knockout. When E2-induced anti-inflammatory responses are attenuated in Sirt3 knockdown or deleted cells, they can conclude as they expect.

Response:

The reviewer addressed an important issue, which we also considered during the study. Indeed, we hardly tried to downregulate Sirt3 with shRNA. Unfortunately, we did not achieve a high KD efficiency with either electroporation or Lipofectamine methods. Additionally, we have tried using Sirt3 inhibitor LC-0296, but we did not observe the expected results in the SOD2 deacetylation.

Since Sirt3 is the major deacetylase in mitochondrial matrix with known anti-oxidative (PMID: 32423865) and inti-inflammatory (PMID: 36786945) effects, one may assume the role of Sirt3 in the E2 effects. Nevertheless, we avoided expressions like “Sirt3-caused effects”, rather presented it as “accompanied effects”.

  1. They started their experiments based on Sirt3 upregulation by E2 alone. However, E2 upregulated only Sirt3? Were any other Sirt proteins not influenced by E2? They need to mention something in this regard.

Response:

We addressed this issue in new Figures F and G where the E2 effects on Sirt1 and Sirt5 are shown. No effects of E2 on these sirtuins could be found.

Minor comments

  1. In Figure 1, They only demonstrated induction of Sirt3 protein at 12 h and 24. Probably, the peak of the protein induction may be between 12h and 24 h. Did they perform the time course of Sirt3 protein upregulation? If so, the peak time needs to be clarified.

Response:

Thank you for your comment. We did not perform a time course of Sirt3 protein expression between 12h and 24h.

  1. In addition to BMM, how about peritoneal macrophages in terms of Sirt 3 induction by E2? They need to mention why BMM was used in comparison with RAW cells.

Response:

We only analyzed BMM and not peritoneal macrophages. We decided to use BMM since they yielded the most macrophages with the best homogeneity (PMID: 29204985, PMID: 23384230). It is not uncommon to compared RAW cells with bone marrow macrophages (PMID: 37211559).

  1. In Figure 2 the legends regarding 2C-E were missing. They need to add them.

Response:

We apologize for this error. We now corrected the figure legend.

  1. As M2 polarity marker, arginase-1 or IRF4 is well known. Contrasting to RELMa, how about the marker induction by E2?

Response:

RELMα is a well stablished M2 marker in mouse macrophages (PMID: 31126996, PMID: 24695852 PMID: 34867999). We did not analyze the E2-dependent effects on RELMα expression.

  1. As shown in Figure 3F, E2-induced Sirt 3 upregulation was confirmed in each M0, M1, and M2 type RAW cell; however, deacetylation was attenuated in M1 alone. They need to discuss this phenomenon.

Response:

The important issue raised by the reviewer has been addressed in the discussion:

“Although E2 upregulated Sirt3 in all types of macrophages, SOD2 deacetylation was observed only in M1 macrophages. The cause of this phenomenon is currently unclear and could be due to differences in Sirt3 activity. Indeed, Sirt3 is an NAD+-dependent deacetylase. Among several mechanisms controlling NAD+ concentration in the mitochondrial matrix, SLC25A51, a recently discovered mitochondrial NAD+ transporter, seems to play an important role in mammals (PMID: 35932995, PMID: 33262325). Although the E2 effects on the SLC25A51 expression remain unknown, the activity of the transporter is regulated by cardiolipin, probably through the direct binding to the three binding motifs at the SLC25A51 (PMID: 38108469). Since E2 may increase cardiolipin concentration in mammalian cells (PMID: 31323169), one may speculate that the E2-cardiolipin-SLC25A51 axis may be responsible for the SOD2 deacetylation in M1 macrophages. Whether this or other mechanisms are responsible for the M1 macrophages-specific SOD2 deacetylation, remains obscure and should be a topic of future studies.”

Reviewer 3 Report

Comments and Suggestions for Authors

In this manuscript, Barcena et al hypothesized that activation of estrogen receptors leads to elevation of the mitochondrial deacetylase Sirt3, which supports mitochondrial function and mitigates the pro-inflammatory phenotype in macrophages. Our study suggests that E2 treatment upregulates Sirt3 expression in macrophages. In primary BMM, female-specific Sirt3 upregulation was observed. The Sirt3 upregulation was accompanied by mitochondrial protein deacetylation and alleviation of the oxidative and pro-inflammatory phenotype in M1 macrophages. Thus, the authos propose that the E2-Sirt3 axis might be used in a therapeutic strategy to fight chronic systemic inflammation and prevent the development of inflammation-linked diseases

Comments:

1)       To confirm the role of Srt3 in estradiol-induced mitochondrial protein deacetylation and alleviation of oxidative stress and pro-inflammatory phenotype, it would be interesting to explore if Sirt3 inhibition can prevent its effect.

2)       Are other axes of mtUPR activated by estradiol?

3)       What it’s the effect of estradiol on bioenergetic parameters in both cellular models? The results of Seahorse assays and all bioenergetic parameters must be shown.

4)       It would be of interest to examine the effect of estradiol on the expression levels of mitochondrial proteins.

5)       It would be of interest to examine the effect of estradiol on inflammasome activation.

6)       Please, review that the Material and Methods section provides information from which companies the compounds are acquired.

Author Response

Comments and Suggestions for Authors

1) To confirm the role of Srt3 in estradiol-induced mitochondrial protein deacetylation and alleviation of oxidative stress and pro-inflammatory phenotype, it would be interesting to explore if Sirt3 inhibition can prevent its effect.

Response:

The reviewer addressed an important issue, which we also considered during the study. Indeed, we hardly tried to downregulate Sirt3 with shRNA. Unfortunately, we did not achieve a high KD efficiency with either electroporation or Lipofectamine methods. Additionally, we have tried using Sirt3 inhibitor LC-0296, but we did not observe the expected results in the SOD2 deacetylation.

Since Sirt3 is the major deacetylase in mitochondrial matrix with known anti-oxidative (PMID: 32423865) and inti-inflammatory (PMID: 36786945) effects, one may assume the role of Sirt3 in the E2 effects. Nevertheless, we avoided expressions like “Sirt3-caused effects”, rather presented it as “accompanied effects”.

2) Are other axes of mtUPR activated by estradiol?

Response:

Although estrogen receptors and Sirt3 play a role in the mitochondrial unfolded protein response, this topic was beyond the scope of the present study.

3) What it’s the effect of estradiol on bioenergetic parameters in both cellular models? The results of Seahorse assays and all bioenergetic parameters must be shown.

Response:

We included these results in Supplementary Figure 3.

4) It would be of interest to examine the effect of estradiol on the expression levels of mitochondrial proteins.

Response:

We also proved whether E2 may affect expression of proteins in the mitochondrial respiratory chain. We included the data in the Supplementary Figure 4.

5) It would be of interest to examine the effect of estradiol on inflammasome activation.

Response:

We analyzed expression of IL-1β and IL-18, which are part of the inflammasome pathway, but could not see any effect of E2. We addressed this issue in Results (Line 324).

6) Please, review that the Material and Methods section provides information from which companies the compounds are acquired.

Response:

We provide information from which companies the compounds were acquired in Materials and Methods.

Round 2

Reviewer 2 Report

Comments and Suggestions for Authors

The Authors responded properly to the referee's comments and concerns and mentioned what is an unsolved issue and is not performed in this study. Based on the limitation of this study, they have revised their expressions. The referee has nothing to mention further.

Author Response

We thank the reviewer for the positive comment.

Reviewer 3 Report

Comments and Suggestions for Authors

1) Please, confirm the implication of SIRT3 on mitochondrial protein deacetylation and alleviation of oxidative stress and pro-inflammatory phenotype using 3-TYP.

2) How do you interpret that estradiol significantly upregulated the expression of complexes I, IV, and V in M2 macrophages without affecting bioenergetics parameters?

My impression is these data are not consistent.

Minor

In 3.2. Estradiol Increases Sirt3 Expression and Mitochondrial Protein Acetylation in Primary BMM from Female but not from Male Mice

It should be Mitochondrial Protein Desacetylation

Author Response

1) Please, confirm the implication of SIRT3 on mitochondrial protein deacetylation and alleviation of oxidative stress and pro-inflammatory phenotype using 3-TYP.

Response:

We greatly appreciate the reviewer’s suggestion to use 3-TYP. However, the laboratory where this research was conducted has since been dissolved, and we no longer have access to the necessary resources to perform these experiments. Given these constraints, we have provided a discussion of the limitations of our study in the section “Limitations of the study”. We believe that the current data, supported by the existing literature, sufficiently addresses the role of SIRT3 in the context of our study.

We hope that these considerations are helpful, and we remain open to any further suggestions the reviewer may have.

2) How do you interpret that estradiol significantly upregulated the expression of complexes I, IV, and V in M2 macrophages without affecting bioenergetics parameters?

Response:

We were also surprised by this phenomenon. However, one should consider that the upregulation of proteins at the mitochondrial ETC does not necessarily increase their activity. In a previous report (PMID: 30051530), we observed an increased expression of cytochrome c oxidase (COX) in H9C2 cells after estradiol treatment, whereas COX activity was rather reduced relative to COX content, suggesting that overall COX activity remained unchanged. We also observed no effects of estradiol on mitochondrial respiration (unpublished data).

Minor

In 3.2. Estradiol Increases Sirt3 Expression and Mitochondrial Protein Acetylation in Primary BMM from Female but not from Male Mice

It should be Mitochondrial Protein Desacetylation

Response:

The correction has been done.

Round 3

Reviewer 3 Report

Comments and Suggestions for Authors

The authors have addressed all my concerns